# Immunopathology of RSV: An Updated Review

**DOI:** 10.3390/v13122478

**Published:** 2021-12-10

**Authors:** Harrison C. Bergeron, Ralph A. Tripp

**Affiliations:** Department of Infectious Diseases, College of Veterinary Medicine, University of Georgia, Athens, GA 30602, USA; harrison.bergeron@uga.edu

**Keywords:** respiratory syncytial virus, RSV, immunity, immunopathology, host–pathogen interaction

## Abstract

RSV is a leading cause of respiratory tract disease in infants and the elderly. RSV has limited therapeutic interventions and no FDA-approved vaccine. Gaps in our understanding of virus–host interactions and immunity contribute to the lack of biological countermeasures. This review updates the current understanding of RSV immunity and immunopathology with a focus on interferon responses, animal modeling, and correlates of protection.

## 1. Respiratory Syncytial Virus (RSV)

### 1.1. RSV Overview

RSV is a member of the *Pneumoviridae* genus and contains a single-stranded non-segmented negative-sense RNA genome approximately 15,200 nt in length [1]. Its genome contains 10 open reading frames (ORFs) which encode 11 proteins. From 3′ to 5′, these genes include two non-structural proteins (NS1 and NS2), two nucleocapsid proteins (N and P), one inner envelope membrane protein (M1), three surface proteins that coat the virion—small hydrophobic (SH), attachment (G), and fusion (F)—M2 which contains overlapping ORFs, resulting in the production of M2.1 and M2.2, and large (L) protein [2]. G and F proteins are the major antigenic proteins. RSV is pleomorphic, i.e., spherical, asymmetrical, and filamentous [3], and categorized into subgroups A and B based on the sequence of the G protein [4,5,6]. RSV subtypes co-circulate and often cause reinfection with the same strain [7].

RSV can be transmitted through respiratory droplets or fomites which infect the upper respiratory tract (URT) via nasopharyngeal or conjunctival mucosa [8]. From the URT, RSV spreads to the lower respiratory tract, primarily infecting polarized ciliated human airway epithelial cells (hAECs) [9], leading to lower respiratory tract infection (LRTI), bronchiolitis, and/or pneumonia [10,11,12]. Along with respiratory epithelial cells, RSV has also been reported to infect CX3CR1^+^ neonatal regulatory B lymphocytes [13], primary neurons [14], alveolar macrophages [15], dendritic cells [16], neutrophils [17], mast cells [18], and T cells [19].

Several host cell receptors for RSV are proposed. The attachment (G) protein is responsible for virus attachment and is a ~300 amino acid glycoprotein consisting of cytoplasmic (CP), transmembrane (TM), and extracellular (ecto) domains [20]. Due to an alternative translation site within the TM (Met48), RSV G protein is membrane bound (mG) and soluble (sG) [21]. sG can be detected as early as 12 h post-infection (pi) and is thought to act as an antigen decoy [22] and induce aberrant immune responses [21,23,24]. G protein contains two heavily glycosylated mucin-like domains that flank a highly conserved central conserved domain (CCD), a CX3C motif, and a heparin-binding domain (HBD) [25]. In immortalized cell lines, the HBD is responsible for substantial binding via cell surface glycosaminoglycans (GAGs) which can be blocked with the addition of exogenous heparin sulfate [20,26]. For hAECs lacking any detectable heparin sulfate or other proteoglycans, other modalities of attachment are required [27]. The conserved CX3C chemokine motif in the G protein binds to CX3CR1, a chemokine receptor found on some immune cells and respiratory epithelial cells [28,29]. CX3CR1 is the receptor for the chemokine CX3CL1 (fractalkine or FKN) [30,31,32,33,34]. Like G protein, FKN is membrane bound and soluble, is glycosylated, and contains HBDs [35]. Recent high-resolution crystal structures suggest conformational epitopes requiring proper folding of the cysteine noose located in the central conserved region, suggesting that G protein binds to CX3CR1 different from CX3CL1 despite functional mimicry [36]. Other studies have proposed that annexin II [37] and Toll-like receptor 4 (TLR-4) may also bind G protein [38].

Once G protein attaches to respiratory epithelial cells, the F protein mediates cell fusion, resulting in viral entry and infection [39,40]. This fusion event is catalyzed by the prefusion F protein binding to the host cell receptor causing a conformational change to a post-fusion conformation, fusing the virion with the host cell, and the formation of the fusion pore [41]. Receptors for this fusion event include nucleolin [42,43,44], and co-receptor candidates TLR-4 [45,46], EGFR [47], ICAM-1 [48], IGFR-1 [49], and C-type lectins [50]. Fusion releases vRNA into the host cytoplasm where the nucleoproteins (N, P, and L) initiate transcription in a 3′ to 5′ gradient fashion. The polymerase transcribes each gene, resulting in 5′ to 3′ subgenomic mRNAs, which are translated into viral proteins by the host cell machinery. To replicate the viral genome, polymerase converts the genome to antisense (5′ to 3′) to be used as the template for generating new negative-sense copies. Genomes assemble with viral proteins in the cytoplasm of infected cells to form new virions. Once formed, virion buds or syncytia are formed with neighboring cells mediated by F protein. Syncytia of epithelial cells leads to pathology through alteration of the airway integrity [51].

### 1.2. RSV Epidemiology

RSV infection causes a substantial disease burden in the infant, immunocompromised, and elderly populations with nearly all children infected with RSV by age two [52,53,54]. It is estimated to cause between 55,000 and 200,000 deaths in children under 5 years of age annually with the most serious disease in infants <1 year of age especially in low-income countries [55,56]. RSV is a leading cause of infant hospitalization and contributes substantially to medical intervention required for the elderly [57,58,59]. Moderate-to-severe RSV disease may also lead to the development of asthma and chronic wheeze later in life, even in children with no atopic predisposition [60,61,62,63]. This phenomenon is potentially mediated by the immune sensitization to RSV, lung development, neuronal development, or a combination of these factors and others [64]. Maternal antibodies may offer some protection to newborn infants; however, these antibodies wane within weeks after birth, and titers of antibodies vary between mothers [65,66]. Of note, during the COVID-19 pandemic, RSV infections (as well as other respiratory viruses included influenza A, influenza B, and adenovirus) fell possibly due to improved hygiene practices, social distancing, and school closures [67,68]. Children in Melbourne, Australia had between 68.8 and 100% reduction in RSV cases, which correlated with the strictness of the lockdown measures in place at that time [67].

Despite the known disease burden, there is no FDA-approved RSV vaccine available. As of July 2019, there were 121 clinical trials evaluating RSV vaccines [69]. In brief, vaccine platforms including virus particle based, nucleic acid, live attenuated, subunit and vector based are being pursued for infants, children, and the elderly as well as maternal vaccines to protect newborn infants [70,71,72]. Many vaccine candidates largely focus on generating pre-F antibodies that are potently neutralizing; live-attenuated vaccines may exclude certain proteins or epitopes while retaining the pre-F conformation [73]. Importantly, live-attenuated vaccines are likely exclusive to the infant population due to a lack of prior exposure while particle-based vaccines may be used for the various populations at risk for RSV disease [74].

The only specific prophylactic countermeasure available is palivizumab (Synagis^®^) which was licensed over 20 years ago [75]. Palivizumab is a humanized monoclonal Ab (mAb) targeting the F protein and is restricted for use in high-risk infants [76]. Palivizumab is administered monthly by injection and reduces hospitalization by ~50%; however, it is not recommended for treatment of wheeze or asthma by the American Academy of Pediatrics [77] and is only approved prophylactic [78]. Next-generation mAb candidates are in preclinical and clinical trials some demonstrating promise as a superior drug, e.g., nirsevimab [79,80]. mAbs including motavizumab and suptavumab (REGN2222) have recently failed to reach clinical trial endpoints [81]. Small- and large-molecule drugs in RSV therapeutic development have recently been reviewed [82].

## 2. The Immune Response to RSV

The respiratory system has a myriad of physical and biochemical features to protect it from agents such as viruses. Respiratory epithelial cells form tight junctions lining the respiratory tract and secrete mucins to help clear pathogens [83,84]. Surfactant-associated proteins (SPs) are members of collectins (collagenous C-type lectins) which reside on the apical surface of the epithelium and assist in the opsonization of potential pathogens [85]. Along with delivering a physical barrier, respiratory epithelial cells may also be phagocytic for pathogens such as bacterial and fungus as these cells also act as secondary phagocytic cells engulfing pathogens [86]. Respiratory epithelial cells are the primary target for RSV infection [87,88].

### 2.1. Innate Responses

Innate immunity uses germline-encoded receptors, i.e., pattern recognition receptors (PRRs) that respond to pathogen-associated molecular patterns (PAMPs) conserved among pathogens [89]. PRR-sensing leads to signaling cascades that induce transcription factors that upregulate antiviral and pro-inflammatory cytokines [90,91]. The antiviral cytokines include type I (IFNα, IFNβ) and type III (IFNλ) interferons (IFNs), whereas type II (IFNγ) IFN promotes immune cell activation [92,93]. Early IFN expression induces IFN-stimulated genes (ISGs) which modify the immune response [94]. Viral replication induces differential kinetics and magnitudes of the host response affecting the outcome of an innate and adaptive immune response. Leukocytes mediate the innate response, and the cytokines and chemokines produced to contribute to host protection from virus-induced pathology.

#### 2.1.1. Neutrophils or PMNs

One of the first innate immune cells to respond to RSV infection are neutrophils or polymorphonuclear leukocytes (PMNs) that include neutrophils, eosinophils, basophils, and mast cells [95]. In one study sampling the bronchoalveolar lavage (BAL) of infants with severe RSV bronchiolitis, neutrophils accounted for a majority (76–93%) of innate immune cells [96]. RSV infection induces interleukin (IL)-8 secretion, a neutrophil chemokine [97,98]. It has been shown that RSV F protein can induce NETosis, a form of cell death characterized by the release of decondensed chromatin and granular contents from neutrophils to the extracellular space associated with binding of TLR-4, thus illustrating a potential mechanism for inflammation-induced by RSV F protein [99]. TLR-4 is canonically associated with lipopolysaccharides (LPS), an endotoxin found on Gram-negative bacteria that results in upregulation of IL-8 [100]. Thus, the initial mediation of TLR-4 by RSV F protein induces a positive neutrophil feedback loop recruiting more neutrophils to the lung and potential immunopathology.

Polymorphisms that result in increased IL-8 secretion are associated with greater RSV disease severity and wheeze [101,102], and neutrophils have been linked to epithelial cell damage [103] and mortality in young children with untreated RSV bronchiolitis [104]. Recent nasal immune profiling of infants with RSV bronchiolitis showed increased levels of IL-8 [105], and global mRNA expression showed increased neutrophil signatures in severe vs. mild RSV infection in infants [106]. Further, transient neutropenia due to young age was not a risk factor for immunocompetent infants to develop serious RSV disease [107]. TLR-2 is also involved in pulmonary neutrophil during early infection (24 hpi) linked to the expression of CCL2 [108] which may be in part driven by RSV G protein [109]. Along with NETosis, neutrophils also produce pro-inflammatory cytokines such as tumor necrosis factor-alpha (TNFα) which may contribute to immunopathology [110].

Early RSV vaccine trials investigated formalin-inactivated RSV (FI-RSV) vaccination of young children that unfortunately resulted in enhanced RSV disease including two infant deaths following natural RSV infection [111,112,113]. Initial reports suggested eosinophilia as driving immunopathology; however, recent analysis suggests that neutrophilia has a major role in enhanced respiratory disease (ERD) [12,114]. Contrary to these findings, one published report in mice suggests that neutrophils do not affect lung virus load or contribute to pro-inflammatory responses following RSV infection [115]. Comparing human responses to those in mice uses different metrics especially when trying to compare mild to severe disease given that mice are only semi-permissive to RSV, it is plausible that neutrophils impact RSV immunopathology.

#### 2.1.2. Alveolar Macrophages

Alveolar macrophages (AMs) are present at the luminal surface of the alveolar lung space and are early responders to lung epithelial insult [116]. RSV infection mouse or human AMs induces TNFα-mediated necrosis mediated through the RIPK1/3/MLKL expression pathways [117]. RSV replication leads to upregulated macrophage migration inhibitory factor (MIF) expression leading to modified cytokine production by AMs [118]. Interestingly, this pathway is associated with increases in the pro-inflammatory TNFα as well as anti-inflammatory IL-10, suggesting that regulation of these cytokines is pivotal in balancing protection and pathology. One such regulator is myeloid PPAR-γ expression that in mice has been shown to reduce inflammatory markers such as TNFα and IL-1β [119]. AMs stimulated by TNFα and monocyte chemoattractant protein 1 (MCP-1) in the allergic airway murine model resulted in increased production of IFNγ and IL-27 [120]. Another regulator, i.e., transforming growth factor (TGF)-β1 disrupts antiviral host responses including type I IFNs during RSV infection in human and murine AMs [121]. AMs have been shown to also express IL-33, a driver of Th2-associated cytokine production, in a mitogen-activated protein kinase (MAPK)-dependent pathway leading to activation of nuclear factor kappa-light-chain enhancer of activated B cells (NF-κB) [122].

Relevant to FI-RSV-mediated ERD, challenging FI-RSV immune mice with RSV results in fewer AMs compared to non-primed or virus-like particle (VLP) F protein primed mice [123]. AMs expressing CD169+ are responsible for the capture of pathogens and are frequently the first cell type infected and thereby provide a confined source of antigen [124]. Interestingly, diphtheria toxin receptor (DTR) transgenic mice depleted of CD169^+^ AM cells had reduced pro-inflammatory BAL cytokines while chemokine levels were not affected, and there was an increase in lung inflammatory cells (monocytes, neutrophils, and eosinophils) following RSV challenge [125]. Analogously, RSV G and/or SH proteins have been shown to reduce chemokine expression (e.g., MCP-1 and MIPs) early after RSV infection with a reduction in non-tissue-resident macrophages occurring during RSV infection [126].

#### 2.1.3. Eosinophils

Pulmonary eosinophilia is a hallmark of ERD in animal models of FI-RSV vaccination [127]. Some studies have linked ERD to RSV G protein sensitization. For example, sensitizing mice with recombinant vaccinia virus expressing RSV G protein (vvG), and not F protein or N protein, followed by RSV infection was shown to lead to substantially increased pulmonary eosinophils [128]. In contrast, mice intranasally infected with an RSV mutant virus lacking the G and SH genes develop pulmonary eosinophilia after RSV challenge indicating that the G and SH proteins are not solely responsible for ERD [129]. Other studies have shown that RSV G protein priming modulates eosinophil trafficking and function [130,131,132], and sublingual administration with the RSV G protein CCD primes mice for pulmonary eosinophilia and results in greater eosinophils compared to FI-RSV priming [133]. Sublingual priming was also shown to prime both pulmonary eosinophilia and neutrophilia in the lung tissue [134].

Eosinophilia and neutrophilia are linked to IL-17, as IL-17 depletion rescues this phenotype, and IL-17 is associated with other respiratory diseases such as asthma [135]. Additionally, the expression of IL-5 has been associated with pulmonary eosinophilia and airway hyperresponsiveness following RSV challenge or vaccinia virus G protein (vvG) priming [136,137]. Consistent with this finding, studies with overlapping G protein peptides showed that the G_184–198_ peptide encompassing the CX3C motif stimulates G protein primed murine splenocytes and peripheral blood mononuclear cells (PBMCs) to express IFNγ and IL-5 [138]. While this may indicate a level of immunopathology, one study showed a protective role for eosinophils expressing IL-5 in RSV clearance [139]. Another study examining IL-5^−^/eotaxin^−^ knockout mice showed decreased mucus and airway inflammation following FI-RSV vaccination and RSV challenge with mice expressing lower IL-4 and IL-13 but increased IFNγ levels where the mice maintained RSV infection for a longer duration [140]. In summary, it is unclear if RSV G protein-induced pulmonary eosinophils are pathogenic or protective.

IL-13 expression, ERD, and pulmonary eosinophilia are mediated by vvG and FI-RSV priming [141,142,143]. sG protein has been shown to mediate pulmonary eosinophilia in mice and drive Th2 responses demonstrating its ability to induce aberrant immune responses [131]. These features may be advantageous to reduce RSV clearance in the host. Importantly, it has been shown that the form of G protein, i.e., sG or mG, and how it is delivered impacts the host response to RSV. For example, intranasal delivery of liposome-encapsulated G protein results in reduced pulmonary eosinophilia compared to G protein delivered without liposome encapsulation [144]. The significance of eosinophils in ERD is questioned by the findings from a study of eosinophil-deficient mice primed with vvG, as ERD as measured by weight loss, clinical scores, Penh levels, and pro-inflammatory cytokines (e.g., IFNs and TNFα) still developed despite the absence of eosinophils [145]. Given these different outcomes, it suggests that eosinophils are multifunctional, and their role remains under investigation particularly given the association of eosinophils and asthma [146].

#### 2.1.4. Natural Killer (NK) Cells

NK cells are innate effector lymphocytes that typically control tumors and microbial infections, and are regulatory cells engaging in interactions with dendritic cells, macrophages, T cells, and endothelial cells [147]. NK cells can limit or exacerbate immune responses, promote and influence inflammatory responses, and NK cells can regulate RSV infection. It has been shown that severe RSV disease in infants correlates with single-nucleotide polymorphisms (SNPs) that increased leukocyte immunoglobulin-like receptor B1 (LILRB1^+^) NK cells [148]. LILRB1^+^ NK cells are noteworthy as the population of these cells in infants is generally low, and the receptor functions as an inhibitor of immune responses. Interestingly, in CD4 knockout mice, NK cells were shown to be recruited to the lung, and disease was reduced after FI-RSV priming and RSV challenge, suggesting that NK cells and disease are inversely correlated in the absence of CD4 T cells [149]. Mice infected with a recombinant RSV expressing IL-18 to enhance NK cell activation had reduced lung viral loads compared to wild-type infection and exhibited biphasic weight loss (days 2 and 6) not observed in wild-type mouse infection [150]. The role of NK cells was substantiated by depletion studies in mice infected with RSV expressing IL-18.

An in vitro study examining antibody-dependent enhancement (ADE) showed RSV co-incubated with suboptimal concentrations of neutralizing antibodies led to ADE and increased lung viral loads, enhanced numbers of NK cells, and increased IFNγ expression by NK cells [151]. The NK cells did not secrete increased perforin, suggesting that they were not directly cytotoxic. This observation is relevant when considering maternal antibodies that are induced by RSV vaccination as the findings imply that the maternal antibodies may prime NK cells; however, RSV vaccination may enhance or diminish the response [152].

The NK cell response has been linked to RSV pathogenesis as NK cell depletion reduced disease [153]. Interestingly, NK cell trafficking and function were shown to be diminished in TLR-4-deficient mice compared to C57BL/10Sn (TLR-4 expressing) mice after RSV infection [154]. This may be linked to RSV F and G proteins both of which are implicated in agonism and antagonism of TLR-4, respectively. Mice infected with wild-type RSV B1 have decreased NK cell infiltrates in the BAL compared to mice infected with RSV B1 lacking G and SH genes (CP52), suggesting a role for G and/or SH proteins in modifying NK cell responses [129]. The neuropeptide, substance P (SP), was shown to also influence NK cell immunity to RSV as anti-SP mAb treatment resulted in increased NK cells and IFNγ expression [155]. Notably, priming mice with vvG did not result in effect pulmonary NK cells while F and M2 priming did [156]. Taken together, the G protein is implicated in modifying the NK response.

#### 2.1.5. Dendritic Cells

Dendritic cells (DCs) are professional antigen-presenting cells (APC) that function to bridge the innate and adaptive immune responses, and constantly scan the environment to present foreign antigens to adaptive immune cells. DCs have two subpopulations, i.e., plasmacytoid (pDC) or conventional (cDCs) [157]. pDCs and cDCs are similar in that they present antigen to T cells; however, pDCs secrete a substantial amount of type I IFNs [158,159]. pDCs produce higher levels of IFNβ via TLR7/MyD88 signaling during RSV infection compared to cDCs [160]. pDC-depletion in mice has been shown to impair the CTL response and result in increased lung viral loads. Human DCs upregulate CD38 expression via the type I IFN response at early time points post-infection [161]. In IFNβ/YFP reporter mice, MyD88 but not TLR-7 is required to induce IFNβ during RSV infection [162]. Taken together, pDCs secrete substantial type I IFNs during RSV infection via the MyD88 pathway.

cDCs have increased IL-4Rα expression that has been correlated with more severe immunopathology in mice and the development of a Th2 biasing response [163]. Since infants have greater IL-4Rα^+^ cDCs compared to older children this feature may correlate to increased disease severity in younger infants. Overexpression of IL-4Rα in murine cDCs leads to enhanced immunopathology similarly to that of neonatal mice following the RSV challenge [163]. During RSV infection, cDCs and pDCs are increased in the lung even weeks after the infection is cleared, and depleting pDCs enhances pathology and alters cytokine responses post-challenge [164,165]. Interestingly, it was shown that the IFNα produced by pDCs act to clear RSV but does not alter the adaptive immune response or subsequent pathology. In neonatal mice, RSV infection poorly induces IFNα production leading to an IL-4Rα-dependent Th2 response and a lack of viral clearance [159]. pDCs also express IL-33, a cytokine implicated in airway inflammation which leads to pathology during RSV infection [166]. Despite the presence of IFNα or IFNβ, neonatal cDCs are unable to upregulate T cell co-stimulatory molecules including CD80 and CD86 during antigen presentation, which reduces CTL cell priming [167]. Another study evaluating human cord blood cDCs showed limited maturation after sensitizing with RSV; however, influenza A virus sensitization resulted in the maturation of the cDCs demonstrating that while infant cDCs are capable of maturation, RSV is a poor inducer of maturation [168]. Interestingly, RSV infection induced a weaker IFNα response compared to human metapneumovirus (HMPV)-infected human blood monocyte differentiated DCs (moDCs) [159]. Another study showed lower IFNα expression in RSV-infected infants and young children compared to healthy adult pDCs, an effect that is potentially mediated by immature cytosolic RIG-I in younger patients [169].

Inhibiting fatty acid synthesis, and thus mitochondrial function, in DCs infected with RSV shifts a Th2/Th17 response to a Th1 response with reduced lung pathology [170]. The KDM6 gene, which codes for a demethylase, is upregulated during RSV infection and this gene activates transcription in DCs, resulting in the production of inflammatory cytokines, chemokines, resulting in pathology [171]. This is likely important in RSV vaccine development. In mice, RSV infection of pDCs that are co-cultured with T cells has reduced the transformation of Th17 cells to FoxP3^+^ Tregs [172]. Similarly in mice, RSV infection of DCs impairs T cell activation by affecting synapse assembly [173]. In humans, RSV infection of cord blood-derived DCs leads to altered cytokine profiles and reduced capacity to induce T cell proliferation and functional responses [174]. Taken together, these studies show that a protective DC response is related to T cell priming, as DC immaturity results in poor IFNα/β responses during RSV, resulting in Th2 cytokine responses.

### 2.2. Adaptive Immune Responses

Adaptive immunity follows innate immune responses. Adaptive immunity is characterized by immunological memory referring to the ability to quickly and robustly respond to previously encountered pathogens in an antigen-specific manner. Adaptive immunity is separated into humoral or cellular immunity. The humoral immune response encompasses B cells and their products, namely antibodies. The cellular immune response includes T cell subsets. Adaptive immunity is necessary for RSV clearance, and the establishment of long-term memory is needed to protect against future infections.

#### 2.2.1. B Cells

B cells express antibodies that act as B cell receptors. When naïve or memory B cells are activated they proliferate and differentiate into antibody-secreting effector cells or plasmablasts [175]. B cells also present antigens and secrete cytokines. B cell activation occurs in the spleen and lymph nodes. Antigens that activate B cells with the help of T cells are known as T cell-dependent antigens, while antigens that activate B cells without T cell help are known as T cell-independent antigens [176]. RSV-infected infants have increased B cells in their peripheral blood mononuclear cells (PBMCs) [177,178], and B cell-activating factor (BAFF) and ‘a proliferation-inducing ligand’ or APRIL is present in the lung epithelium that correlates with IgA and mucosal antibody responses [179]. Similarly, a study of RSV-infected mice showed increased BAFF and B cell chemoattractant CXCL13 expression in the lung [180].

Concerning antibodies expressed by B cells, anti-F protein antibodies from the adenoids of young children have been shown to have high binding affinity and greater RSV neutralization compared to peripheral blood [181]. Anti-F protein antibodies from infants infected with RSV shows somatic hypermutations (SHM) and immunoglobulin (Ig) class switching that increases with age [182]. Notably, young infants have an immature SHM function that limits the antibody repertoire. Importantly, the most common V-gene antibodies pairs against F protein are VH3-21:VL1-40 or VH3-11:VL1-40 which targeted site III, are neutralizing, and do not require SHM. Another study showed post-fusion F protein vaccination induces pre- and post-F memory B cells and the antibodies were all neutralizing to some degree [183]. Importantly, RSV can infect neonatal B cells via CX3CR1 leading to increased pathology, secretion of IL-10, and an increase in the Th2 response [13]. It was recently discovered the B cell response to RSV is linked to type I IFN receptor expression response, and RSV is known to reduce robust IFN responses potentially translating to reduced B cell function in newborns [184]. ERD mediated by FI-RSV priming is also associated with low-affinity, non-neutralizing antibody responses further demonstrating importance of antibody function and epitope in the context of immunopathology [185].

#### 2.2.2. T Cells

T cells express a T cell receptor (TCR) on their cell surface and belong to two major subpopulations, i.e., CD8+ cytotoxic T cells (CTLs) or CD4+ helper T cells. CTLs can directly kill virally infected cells, and T helper cells assist other lymphocytes including maturation of B cells into plasma cells and memory B cells, and activation of CTL and macrophages [186]. CD4+ T cells can secrete a myriad of cytokines that contribute to B cell stimulation, antibody proliferation and class switching, effective CTL responses, and innate cell activation [187]. CD4+ T cells can differentiate into one of several subtypes each of which has different roles. For example, Th1 cells are characterized by their expression of IFNγ and Tbet producing an inflammatory response [188]. Th2 cells are characterized by expression of IL-4 and drive differentiation and antibody production by B cells [189]. Th17cells express IL-17 and have a role in gut and mucosal defenses [190]. Th9 cells express IL-9 and defend against helminths [191], and Tfh express IL-21 and IL-4 providing B cell help [192,193]. Naive T cells can expand and differentiate into memory and effector T cells after they encounter their cognate antigen in the context of a major histocompatibility complex (MHC) [194]. There are several memory T cell subtypes including central memory T cells (CD45RO+, CCR7+, and CD62L), effector memory T cells (CD45RO+, CCR7-, CD62L-), and tissue-resident memory T cells (CD103+) [195,196].

#### 2.2.3. CTLs

Studies depleting T cells in RSV-infected mice indicated the role of CTLs in controlling RSV and concomitant immunopathology [197]. In adults, it has been shown that resident memory CTL (Trm) proliferate extensively to RSV infection compared to the circulating T cells [198]. Interestingly, this study showed that the Trm were phenotypically lacking cytotoxic markers and had reduced production of pro-inflammatory cytokines but were correlated with protection from RSV disease. Consistent with the lack of cytotoxic function, perforin-depleted mice infected with RSV cleared RSV similarly as wild-type mice likely through Fas/FasL and pro-inflammatory cytokine expression, specifically TNFα [199]. TNFα may also contribute to lung pathology [200]. In mouse studies that examined CTL cell DNA vaccination with an RSV M2 gene, it was shown that DNA vaccination led to a non-protective, pathological response mediated by IFNγ and TNFα secreting T cells [201]. This study also evaluated the passive intranasal transfer of splenic CTLs from RSV-sensitized mice to naïve mice that resulted in protection, reduced RSV lung titers, and increased the IFNγ response post-RSV challenge.

The route of RSV vaccination can affect immunity and disease pathogenesis. For example, when murine cytomegalovirus (MCMV) expressing RSV M protein was administered intranasally (i.n.) as opposed to intraperitoneally (i.p.), disease pathology was ameliorated, and a robust lung-specific T cell response followed [202]. Interestingly, BALB/c mice intravenously (i.v.) vaccinated with an H-2k^d^ restricted RSV M2 antigen generated greater protective polyfunctional CTLs compared to intraperitoneally by approximately 4-fold [203].

RSV G protein has a single MHC-I H-2L^d^ restricted epitope [204]; however, RSV G protein modifies CTL responses. For example, one study examining RSV G_169–198_ peptide nanoparticle vaccination showed increased IFNγ producing MHC class I H-2K^d^ restricted M2-specific CTLs following RSV challenge, an effect linked to CX3C–CX3CR1 interaction [205]. Interestingly, including a G protein CX3C motif in an influenza vaccine was shown to enhance anti-influenza CTL responses [206]. This may be because the G protein CX3C motif affects CX3CR1^+^ CTL trafficking to the lung mediating a Th2-type response [207].

#### 2.2.4. CD4+ T Cells

Th cells function to help other immune cells. For example, they help activate B cells to produce antibodies, feedback to promote CTL activity and function and provide cytokine production which stimulates and activates immune cells [194]. The immunopathology of RSV infection is typically described in the context of Th1- or Th2-type responses. The canonical cytokines associated with Th1 cells are IFNγ and IL-2, while for Th2 cells often IL-4, IL-5, and IL-13 are used. Th1 cells defend the host against intracellular parasites including viruses; Th2 cells defend against extracellular parasites such as helminths. Th1 cells produce IFNγ that stimulates macrophage activation [208]. IL-4 results in IgG1 and IgE class switching [209], IL-5 is a potent eosinophil maturation molecule [210], and IL-13 affects mucus production and is also implicated in asthma [211]. The presence of Th1 cytokines downregulates Th2 cell activation and vice versa. The FI-RSV vaccine primes for Th2-type responses and is a comparator of RSV vaccine safety [212,213]. Further, infants with severe RSV disease have a higher ratio of Th2/Th1 cytokines compared to those having mild disease [213,214,215,216].

The RSV G protein influences Th2-type immune responses, and immune dysregulation is linked to CX3C–CX3CR1 interaction and sG expression [129,217,218,219]. The RSV G protein has been shown to prime for ERD in animal models [12,217]. This feature has hindered RSV vaccine development particularly for vaccine candidates involving the G protein. Accumulating evidence has shown the protective benefits of G protein-mediated antibodies specifically regarding disease pathogenesis, thus G protein-based vaccines are being reconsidered [220]. Studies in mice have suggested that modification of the CX3C motif [221,222] and/or CCD [223] could elicit a safe and protective response and shift Th2-type responses to Th1-type or balanced responses. A G protein DNA vaccine (pVAX1/3G_148–198_) was shown to induce a Th1-type biased response and protect mice from the disease [224]. Additionally, mice were inoculated with a temperature-sensitive (ts) live-attenuated influenza HA/G protein CCD vaccine and this provided protection against influenza and RSV [225]. Thus, the G protein offers an opportunity to induce protective immunity.

Various adjuvants have been tested to improve RSV vaccine candidates or to induce a Th1-type or more balanced Th responses. Vaccination studies in mice using alum and pre-fusion F protein led to ERD compared to a balanced Th1/Th2 response using the adjuvant Advax-SM, a plant-based polysaccharide adjuvant [226,227]. Using an adenovirus serotype 26 (Ad26) vaccine platform and the pre-fusion F protein, a Th1-type immune response was induced in vaccinated mice which lacked detectable immunopathology [228]. However, suboptimal vaccine dosing of pre- or post-fusion F protein caused ERD despite adjuvanting with a Th1-biasing TLR-4 agonist. ERD was prevented in mice optimally vaccinated using doses that induced a robust neutralizing antibody response demonstrating the importance of not only the adjuvant but proper dosing [229]. A multiplex vaccine containing both RSV F and G proteins, i.e., SBP-FG [230] and VLP-FG [231] resulted in a protective Th1-type response. Another multiplex vaccine called G1F/M2 incorporates a G protein neutralizing epitope, and an M2 CTL epitope, that is adjuvanted with CpG2006 (a Th1-biased TLR-9 agonist) was shown to induce a Th1-type response with reduced pulmonary inflammation and disease in RSV-infected mice [232]. Additionally, a recombinant baculovirus expressing a G protein fragment and M2 (Gcf A/Bac M2) was shown to induce an efficacious and protective Th1 response [233], and a multiplex vaccine encompassing the F, N, and M2-1 proteins on a chimpanzee adenovirus vector (PanAd3-RSV) offered protective, robust Th1-type responses [234]. Interestingly, vaccination of mice by microneedle patch using FI-RSV and TLR-4 adjuvant-induced a safe and effective response having reduced Th2-type and ERD responses post-RSV challenge [235]. Interestingly, a soluble F protein vaccine induced a Th2-type response characterized by high levels of IL-4, IL-5, and IL-13 in the BAL, while VLP-F induced a Th1-type response characterized by reduced Th2 cytokines and an increase in IFNγ [236]. Vaccination strategies that target the RSV F and G proteins may be beneficial as anti-bodies specific to this major cell surface viral proteins should protect by virus neutralization, reduce the chances of vaccine escape, and prevent disease pathogenesis, but vaccine platforms, adjuvants, dosing, and their delivery should be rationally developed.

#### 2.2.5. Tregs Cells

Tregs cells are a subclass of CD4^+^ T cells expressing both the CD4, CD25, and the nuclear transcription factor Forkhead box P3 (FoxP3) which determines Treg development and function [237]. FoxP3 is crucial for maintaining the suppression of the immune system. Depleting Treg cells reduces functional B cells that may lead to increased disease in the mouse model [238]. Infants with RSV display reduced numbers of activated Treg cells in the periphery and the lower Treg cytokine (IL-17A, IL-1β, and IL-23) expression that is inversely correlated with disease severity [216,239]. In RSV-infected infants, the Treg responses in PBMCs are altered having decreased IL-2 and Foxp3 compared to healthy controls [236]. Another study showed NS1 suppressed Tregs while NS2 increased levels, suggesting that these proteins function independently to modulate the Treg response [234].

An FI-RSV study in mice showed Treg reduction, which correlated with severe disease was reversed by administering CCL-17 and CCL-22 to recruit pulmonary Tregs [240]. The RSV NS1 protein has been shown to antagonize immunity in part by reducing Treg cells while increasing Th2-type/Th17 cells [241,242]. RSV NS1 protein also increases Th2-type biasing of OX40L, and mAb treatment against OX40L increases Treg cells [243]. NS1 protein has also been shown to increase p-mTOR that reduces Foxp3 expression and Treg cell differentiation [241]. Treg cell depletion has been shown to delay RSV clearance and delay CTL trafficking to the lung [244]. Moreover, depletion increased the pro-inflammatory functions of the CTLs, suggesting that the Treg response is key for mediating the CTL response to clear RSV and reduce CTL-induced immunopathology. In a study that examined RSV-infected Treg-depleted mice depletion was shown to lead to improved lung virus clearance; however, depletion was associated with increased markers of enhanced disease including severe weight loss and BAL cellular influx [245]. Interestingly, this study also showed that pulmonary Tregs cells expressed granzyme B, suggesting a role in controlling RSV infection. In another study examining Treg cell-depleted mice, it was observed that depletion was associated with highly functional Th2-type pulmonary CD4 cells and enhanced pulmonary disease [246].

### 2.3. Interferons (IFNs) and Inteferon Stimulating Genes (ISGs)

IFNs impede viral replication and regulate RSV infection. IFNs are categorized into three types, i.e., type I, II, and type III IFNs. Type I IFNs include IFNα and IFNβ with at least 14 distinct isoforms of IFN-α [247]. pDCs are chief producers of type I IFNs that also include IFNs ε, κ, τ, μ, ζ, ω, and υ; however, these are not as well described. Type II IFN mediate T cell responses and activate macrophages [248]. IFNγ downregulates Th2-type cells and upregulates Th1-type cell responses. An infant cohort study showed RSV disease and wheeze associated with reduced IFNγ and a Th2-type cytokine propensity [249]. IFNγ is a canonical Th1-type cytokine. Type III IFNs include four isoforms of IFNλ (i.e., IFN-λ1, -λ2, -λ3, and -λ4) and are similar in function to type I IFNs [250].

IFNs bind their receptor and induce differential gene expression of interferon-stimulated genes (ISGs), resulting in an amplified antiviral response. Type I IFN binds to the ubiquitous heterodimeric membrane-bound receptor IFN-α/β-R1/2 (IFNAR) [251]. Binding results in activation of NF-κΒ, cAMP-response element-binding protein (CREB), and signal transducer and activator of transcription proteins (STAT). IFNγ binds to the ubiquitous receptor IFN-γR-I and -II. This binding results in the activation of the JAK-STAT pathway [252]. Type III IFNs bind to a complex of IL-10Rβ/2 and IFNLR1 that are found on subsets of epithelial cells and neutrophils, and this binding event cascades to activation of STAT and CREB [253]. These IFNs have distinct and overlapping roles in the context of RSV.

#### 2.3.1. Type I IFN

The SOCS family of proteins are negative-feedback inhibitors of signaling induced by cytokines that act via the JAK/STAT pathway [254,255]. SOCS expression is induced by TLR-4. During RSV infection, early upregulation of SOCS3 results in downregulation of IFNα expression that is linked to RSV F and G protein induction of TLR-4 [256]. IFN signaling is tightly controlled by SOCS expression. Studies have shown that G protein expression modulates SOCS3, and reduces IFNβ levels likely through reduced expression of the ubiquitin-like protein ISG-15 [257]. Other studies haves shown a reduction in IFNβ is due to the blocking of membrane-bound TLR-3 and TLR-4 via sG antagonism [24]. Importantly, anti-G protein mAb that blocks CX3C–CX3CR1 interaction results in increased type I IFN expression, which correlated with reduced pathology [220]. Taken together, these studies show that RSV surface proteins can modulate type I IFN responses through membrane signaling.

NS1 and NS2 proteins disrupt innate immunity and type I IFN levels. Several studies have shown the RSV NS1/NS2 are involved in immune dysregulation during virus replication. It has been shown that IFN levels remain surprisingly low during RSV infection despite viral replication [258]. Type I IFN expression is substantially increased when cells are infected with an RSV deletion mutant lacking NS 1 and NS2 (RSVΔNS1/2) [259]. Several RSV proteins have been shown to suppress both the production and signaling of type I and III IFNs by counteracting host innate signaling proteins, and several ISGs are affected by NS proteins [258,260]. For these reasons, several attenuated vaccine candidates contain deleted NS genes. Infection of primary bronchial epithelial cells with RSVΔNS1/2 was shown to have reduced replication efficiency, suggesting a role for NS1/2 in replication [10]. Interestingly, neonatal mice treated with IFNα before the RSV challenge had increased B cells and IgA levels that correlated with an improved disease outcome [261], and infants with severe RSV had lower type I IFN gene expression compared to mild disease [262]. Taken together, type I IFNs may be protective during RSV infection; however, proteins including NS1, NS2, G, and F act to reduce type I responses. Intriguingly, NS1 has been shown to increase miR-29a levels that are known to decrease type I IFNAR expression, suggesting a means by which NS may act to quell the antiviral response [263].

#### 2.3.2. Type III IFNs

Type I IFN receptors (IFNAR) are ubiquitous, whereas type III receptors are expressed principally on epithelial cells [264]. Type III IFN is thought to mediates a less efficient antiviral response compared to the type I IFN [265]. It is thought that the host reserves the more potent type I IFN response to be used if the local type III IFN response is insufficient [266]. Infants infected with RSV have increased IFNλ levels in nasal specimens that correlate with viral titers [267] and increased respiratory rates [268]. In primary nasal epithelial cells, RSV infection is associated with increases in IFNλ but not type I IFN which is facilitated through the RIG-I pathway [269]. Increasing IFNλ through blockade of prostaglandin D2 (PGD2): PGD2R2 has been shown to improve disease outcomes in neonatal mice and assist in lung viral clearance [270]. Recent studies suggest UV-inactivated RSV, and RSV F protein mediate an IFNλ blocking mechanism via epidermal growth factor receptor (EGFR) agonism [271]. F protein agonism of EGFR is linked to abundant mucin production [47]. Interestingly, RSV replication requires Rab5a involvement in intracellular membrane trafficking, and Rab5a downregulates the IFNλ response in the airway [272]. The RSV G protein modifies IFNλ responses, and infection of Calu-3 cells with a modified RSV having a G protein CX3C region ablation increased IFNλ expression indicating that CX3C–CX3CR1 interaction signals to modify the type III IFN response [273]. Additionally, the RSV G protein upregulates miRNA let-7f [274], and it is believed that this miRNA functions to reduce IFNλ translation. Thus, several RSV proteins may affect type III IFN responses to facilitate viral replication.

## 3. Animal Models

Animal models are used as a surrogate to better understand human responses to RSV infection, but few models are appropriate, as most are semi-permissive to RSV replication except chimpanzees [275,276,277]. Some studies have examined related pneumoviruses, e.g., pneumonia virus of mice (PVM) [278] and bovine RSV (BRSV) [279] as alternate virus models to RSV as these viruses have permissive animal hosts.

### 3.1. Mice

The most widely used species used in RSV studies is the inbred mouse; however, permissiveness to infection varies by strain by up to 100-fold [280]. Mouse strains have varying sensitivity and a vast majority of studies utilize the BALB/c mouse which are biased towards a Th2-type response and are H2^d^ MHC restricted [281]. Interestingly, older mice have shown increased susceptibility to RSV infection [282]. Correlates of immunity and disease pathogenesis in mice can be assessed; however, weight loss is not typically observed in human disease, and does not always correlate with lung damage, lung function, or BAL infiltrates [283,284], thus, mice do not fully recapitulate human disease. Humanized mice have been investigated and in one study, human immune system (HIS) mice were generated from the NOD SCID gamma (NSG) mouse which possessed functional T cells and B cells and challenged with RSV line 19F [285]. Compared to NSG mice, HIS mice lost significant weight and had significantly more lung histopathology with increased BAL cell infiltrates, and increased IL-1β and MUC5ac, and reduced CCL3. Humanized mice appear to be a useful model but are laborious to develop and too costly. An alternative may be to challenge neonatal mice with the rA2-19F that closely mimics RSV disease symptoms in human infants [286]. rA2-19F is a chimeric RSV in which the F protein from the A2 strain was replaced with the F protein from the Line 19 strain (rA2-19F) and infection of adult mice with rA2-19F has been shown to mediate higher lung viral loads compared to either of its parent strains (A2 or line 19) [287]. This is likely a result of five mutations within the F protein which are involved with increased susceptibility to mouse lung, a reduction in type I IFN responses, and increased mucosal responses (including IL-13) [287]. rA2-19F infection has been shown to cause lung pathology, mucin secretion, airway hypersensitivity, and Th2 responses which are diminished via prophylaxis with anti-G protein mAb (131-2G) [287,288].

### 3.2. Cotton Rat

The cotton rat (CR) (*Sigmodon hispidus*) model has been used to justify the advancement of palivizumab in human clinical trials [289,290]. CRs are sensitive to RSV [291] and contain a functional Mx gene set which is important for mounting the innate antiviral response in humans which is absent in many inbred mouse strains [292]. However, the features of the immune response vary as CRs age [293,294]. CRs demonstrate a sex-dependent response to RSV with females showing increased airway hyperresponsiveness and pulmonary edema compared to male littermates [295]. Interestingly, the BAL fluid of CRs is primarily comprised of eosinophils, and challenge with RSV does not increase these cell numbers [296]. Moreover, CRs can be used in maternal antibody studies, as RSV antibodies transfer from dames to pups via the placenta and breast milk [297,298,299,300]. Further, CX3CR1 is a receptor used by RSV for infection of CRs [301]. While the cotton rat model has been used to evaluate drugs and vaccines, limitations of this model include cost, procurement (i.e., limited suppliers), and lack of robust tools to evaluate immune responses.

### 3.3. Non-Human Primates (NHPs)

NHPs are used as models because of their close evolutionary lineage to humans. RSV was first identified in chimpanzees and consequently named chimpanzee coryza agent (CCA) [302]. Apart from chimpanzees, other NHP species are only semi-permissive to RSV infection illustrating the specificity of RSV to humans. Nevertheless, NHPs have been used in several preclinical studies of drugs and vaccines. For example, the RSV fusion inhibitor, TMC353121, was tested in African green monkeys (AGMs) followed by RSV challenge which was shown to reduce lung RSV titers [303]. The RSV vaccine candidate, DS-Cav1-I53-50, is a stabilized pre-F protein nanoparticle vaccine that was evaluated in rhesus macaques and found to be highly immunogenic [304]. Another RSV nanoparticle vaccine candidate, i.e., pre-F-NP, a modified pre-F stabilized protein fused to ferritin nanoparticles also showed increased immunogenicity compared to the DS-Cav1 in Cynomolgus macaques [305]. Trivax, a candidate RSV vaccine that includes three RSV F protein neutralizing epitopes (i.e., 0, II, and IV) was evaluated in naïve and RSV-experienced AGMs [306]. Trivax was shown to be immunogenic in naive and AGMs sensitized to the RSV F protein. In addition, vectored virus vaccine candidates have been evaluated in Cynomolgus macaques, and importantly delivery via i.n. route resulted in broad and robust anti-F IgA responses [307]. A codon-deoptimized live-attenuated vaccine delivered to AGMs generated robust immunity and displayed minimal viral shedding, a point of interest in live-attenuated vaccines [308]. It is of note that many of the NHP studies evaluate responses against a common target, i.e., pre-F, yet differ in platform, adjuvant, and/or delivery.

The use of small animal models such as BALB/c mice aids immunological research and disease pathogenesis studies especially with regard to markers of ERD (i.e., Th2-type cytokines and chemokines), as mice have robust commercially available tools such as antibodies. However, mice are only semi-permissive to RSV and do not recapitulate human disease. CRs are useful for studies determining RSV transmission, including dam-to-pup transmission, as RSV is permissive in this model. However, similar to the mouse model, RSV disease is not recapitulated in the CR and moreover there are limited commercial tools available for studying CRs. NHPs, while requiring more resources than small animals, provide a good model for late preclinical vaccine and/or therapeutic evaluations, particularly when evaluating safety given their close lineage to humans. Thus, while these three animal models individually serve a niche in RSV research, no single animal model can fully serve as a surrogate for human RSV disease. The lack of an all-around animal model has hindered RSV vaccine and therapeutic development.

## 4. Correlates of Protection

A major hurdle in establishing countermeasures to RSV is limited correlates of protection (CoPs) needed to evaluate vaccines and therapeutics. CoPs are determined when there is a direct cause–effect relationship between a factor such as vaccination or treatment and an endpoint. Typically, virus-specific neutralizing antibody titers known to protect against RSV disease are used. Several studies have examined RSV-specific polyclonal or monoclonal antibodies responses associated with neutralizing activities as CoPs [80,309,310,311]. For example, antibody neutralization has been demonstrated by analyses of maternal sera [309,312], cord blood [313], breast milk [314], infant sera [309], and by passive antibody transfer studies [315]. CoPs for RSV remain under investigation and include specific immune responses, viral load, duration of infection, and microRNA responses [316]. One means of evaluating a CoP is to determine the cytokine/chemokine profiles of lung leukocytes and the Th cell subpopulations after sensitization [216]. In general, lower levels of Th2-type cytokines (e.g., IL-4) have been suggested to be associated with reduce disease [317], while MUC5AC expression has been correlated with more severe disease [318] and CXCL4 expression correlated with reduced disease severity [319]. Additionally, MIP-1α a potent eosinophil chemoattractant, has been positively correlated with severe disease in hospitalized infants, and interestingly RSV G protein may inhibit early MIP-1α expression in mice [126].

Along with Th1- or Th2-type polarization, there is macrophage polarization, where macrophages adopt different functional programs in response to the signals from their microenvironment. This ability is linked to their roles being effector cells in the innate immune response, and also removal of cellular debris, embryonic development and tissue repair [320]. Macrophages can be polarized into classic (M1) or alternatively activated (M2) groups based on in vitro studies, in which cultured macrophages were treated with molecules that stimulated their phenotype switching to a particular state [321]. M1 macrophages have been described as pro-inflammatory and important in directing host-defense against pathogens including phagocytosis and secretion of pro-inflammatory cytokines and microbicidal molecules. M2 macrophages have been described in the regulation of the resolution of inflammation and the repair of damaged tissues. M1 macrophages are induced by Th1-type cytokines and TLR-4 interaction, while M2 macrophages are induced by Th2-type cytokines [322]. The role of macrophage polarization and activation during RSV infection is significant. In the asthmatic murine model, TLR-4 signaling linked to by RSV induced M1 activation and reduction in Th2-type cytokines [323]. Other studies have shown an opposite function where M2 macrophages in the lung of RSV-infected mice contribution to Th2-type cytokine production mediated by STAT6 [324,325] contributing to a protective role in RSV infection [326]. For example, increasing M2 macrophage polarization by IL-4/anti-IL-4 immune complexes or PPARγ agonism has been shown to result in decreased lung pathology during RSV infection in mice [327]. Additionally, inhibition of cyclooxygenase-2 (COX-2) in RSV-infected CRs has been shown to increase M2 responses and result in decreased lung pathology [328]. Thus, while cytokines are relevant in RSV disease or protection, macrophage polarization may serve as relevant CoPs.

Another consideration when establishing CoPs is viral load and duration of infection. Traditionally, reduced viral load and a shorter duration of infection are indicative of moderated disease. Sterilizing immunity following RSV infection might likely prevent RSV disease; however, sterilizing immunity is not likely achievable given the many features of RSV to avoid immunity [12,329,330]. Reduced viral loads alone do not correlate with protection from disease as higher viral loads early after infection are correlated with protective innate immunity in infants [331] and elderly patients [332] hospitalized with RSV disease. In addition, antibody levels to RSV have been used to determine CoPs; however, the type of antibody induced and to what antigen (F or G protein) are important. In one study, infants with high levels of maternal antibodies in cord blood were protected from RSV during the first months of life [309], yet the type of antibodies elicited dictate the durability of protection as neutralizing maternal antibodies wane quickly with particular types of antibodies having varying half-lives [333]. Importantly, neutralizing antibody assays generally do not detect non-neutralizing, Fc-mediated responses that may also be protective. Mechanisms by which these responses are protective include antibody-dependent cytotoxicity (ADCC), antibody-dependent cellular phagocytosis (ADCP), and complement-dependent cytotoxicity [334]. However, the antigen–antibody complex associated by Fc and complement binding has been shown to result in ERD (and potential Th2-type biasing) in FI-RSV primed mice [335].

Original antigenic sin (OAS) is the tendency of the immune response to favor responses to an original antigen over a related or new antigen. OAS favors immunologic memory to the original antigen rather than that to new antigens leading to hierarchal responses. OAS can affect the memory response to the original infecting strain or vaccine antigen, accordingly OAS is important in vaccine development because vaccine efficacy could change upon the appearance of a new strain. Thus, the first RSV infection in the infant imprints immunity and may determine the outcome of disease particularly after reinfections. How OAS affects the original response to the infecting strain and subsequent infections needs to be better understood, particularly in RSV vaccine development. Importantly, OAS is an important caveat when evaluating vaccines and therapeutics in naïve animal models [336]. Thus, CoPs determined in the naïve animal models may not align with the markers of OAS-linked immune responses.

MicroRNAs (miRs) have been investigated as biomarkers that may serve as CoPs [337]. In one study, miR profiles expressed in mice vaccinated with different RSV vaccine candidates showed that miR regulation correlated with disease or protection [338]. The G protein CCD was shown to specifically downregulate let-7f and miR-24 [273]. Consistent with miR regulation, HEp-2 cells infected with RSV have downregulated levels of let-7f expression [339] and downregulated levels of miR-140-5p expression which was correlated with disease severity and linked to enhanced inflammation and decreased TLR-4 expression [340]. It has been shown that miR profiles may behave as biomarkers to determine disease severity as downregulation of miR-34b/c-5p has been associated with mucus secretion through reduced regulation of MUC5AC [341].

Importantly, RSV typically infects the very young and old but can infect others including the immune compromised [342] thus standardizing CoPs is complex as there is a wide range affecting outcomes. For example, infants have immunologic and anatomical immaturity, and may not be sensitized to RSV infection. Additionally, their lungs vary in architecture and disease susceptibility [343], and the infant immune system is biased towards a Th2-type response to prevent inflammatory damage to the lungs [344]. As described, Th2-type responses including increased IL-4Rα on cDCs is associated with increased disease in this population. Moreover, infants with genetic mutations for TLR-4, IL-4, IL-8, IL-13, and IL-4Rα are at greater risk for severe disease [102,345,346]. Hospitalization is associated with younger age so by age 5 into adulthood, RSV infections are generally mild, suggesting that the maturation of the immune system, memory T and B cells, and improved antiviral inflammatory responses (i.e., Th1-type) are considered protective and less pathogenic. In contrast, the elderly have a waning immunity and a reduced thymus size which can modify T cell responses, and often have comorbidities or lifestyle choices (i.e., cigarette smoking) that can complicate RSV disease [347]. Interestingly, while RSV disease presents differently in the elderly compared to the infant population [348], elderly immunopathology is similar to infants in that the immunosenescence results in increased Th2-type and Treg and reduced Th1-type and Tmem subset responses [349].

Despite the challenges with elucidating CoPs, Th2-type responses, IFNλ, and eosinophilia and/or neutrophilia are generally associated with immunopathogenesis while balanced and/or Th1-type responses, type-I IFNs and pDCs are considered protective.

## 5. Discussion and Conclusions

RSV causes substantial respiratory tract disease burden particularly in infants and the elderly. Unfortunately, there is no approved vaccine or post-exposure therapeutic available and prophylactic intervention is limited. RSV infection causes immune dysregulation and disease pathogenesis. B cells producing antibodies may neutralize RSV replication, and T cells such as CTL, Th, Treg, and Tmems are important for the resolution of infection. CTLs may also induce pathology, and CD4 Th1 or Th2 cells contribute to disease. Specifically, Th2-type cytokines are thought to be less protective and more pathogenic so a better understanding of the role of cytokines and chemokines, especially IFNs, is essential. While type I IFNs may protect from disease, RSV has various mechanisms to modify the type I response. RSV proteins can reduce antiviral IFNs, and IFNλ levels are key for controlling lung titers and disease severity. Finally, the lack of a reliable, reproducible, and cost-effective animal model has complicated RSV research and disease intervention strategies. Our knowledge of correlates of protection and biomarkers of immunity and disease are currently incomplete; however, new biomarker understanding should deepen our understanding.

Understanding host–pathogen interactions is vital to developing countermeasures to address unmet needs and disease burden. As viral load is not predictably correlated with disease, vaccine and therapeutic countermeasures should consider preventing both virus replication and the proinflammatory response to disease. Specifically, multivalent approaches that target both the G and F proteins could improve disease outcomes. Anti-F protein antibodies have shown clinical efficacy, e.g., palivizumab and nirsevimab, while G protein vaccine strategies have shown promise beyond proof of concept [350,351]. Vaccine candidates encompassing various epitopes such as live-attenuated and codon-deoptimized platforms may be advantageous [221,352]. Host-targeting drugs [353] or immunotherapies, e.g., IFNα [354] that interfere with the virus life cycle and/or improve the immune response may also potentially reduce disease burden with reduced risk of viral escape. Reverse vaccinology and rational structure-based antigen design may yield previously unobtainable responses by designing immunogens to induce potent and protective antibody responses.

## Data Availability

Not applicable.

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
