# Peer review of "Immunopathology of RSV: An Updated Review"

_viruses, 2021, doi:10.3390/v13122478_

Round 1

Reviewer 1 Report

Immunopathology of RSV:  An update review.

Nergeron HC and Tripp RA

This is a comprehensive and well written review from the laboratory of a very experience senior investigator in the field of respiratory syncytial virus (RSV) infection and disease.  The review recounts accurately the cellular mediators and cytokines/chemokines involved in protection and disease, both in infants and adults and it is also paralleled with observations in different animal models. 

There are two points that would add to the discussion on correlates of protection.  First, it would be important to emphasize that there are significant immunological factors, many time overlooked, that are relevant and unique to each of the different circumstances surrounding human RSV infections that should be defined more clearly in special animal models (no naïve ones).  A big one is the pre-existent immunity to RSV (either due to the circulation of maternal antibodies or due to previous RSV infections) which has been started to be looked at, although not presented in this review.  These conditions change completely the landscape of infection and disease and are commonly not introduced in the mix of variables in experimental animal models.  The second is to recount previous observations of the role of alternative activated macrophages (or M2 macrophages) in the resolution of RSV lung pathology, that has been previously described in murine models (in which PPARg also play a pivotal role) and would be important to include to broaden the reader’s view.

Author Response

We appreciate the reviewer’s time and efforts, and have addressed the reviewer’s suggestions as highlighted in the revised manuscript.   

Reviewer 2 Report

This review on the immunopathogenesis of RSV infection by Bergeron and Tripp provides to date, comprehensive, and in-depth summary of the current knowledge on his topic. It is well-written and very detailed on the role of the individual immune cells in the pathogenesis of RSV and for enhanced disease that has been seen following vaccination with formalin-inactivated RSV. There are only minor suggestions:

Vaccines are only mentioned briefly (lines 83-83). It may be helpful to provide a little more detail on promising vaccine candidates (e.g. recombinant RSV, prefusion F). Later in the description of the animal models, some of the vaccine candidates are mentioned. It would be helpful if this would be put in context of each other.

It is stated that RSV infection may also lead to the development of asthma and chronic wheeze later in life (lines 76-77). This statement seems a little superficial, and the reference quoted for that is not optimal to support this statement.

Reviewer 3 Report

Review of review manuscript Immunopathology of RSV: An Updated Review

The manuscript entitled „ Immunopathology of RSV: An Updated Review" from Bergeron and Tripp reviews the immunopathology induced by the infection with RSV.

Overall, the presented manuscript is very well-written and it is a comprehensive review that covered the current state of knowledge on immune response and pathogenesis upon RSV infection, and addressed critical aspects in RSV vaccine development.  

Major points:

In general, it is clear that not all aspects of RSV mediated immunopathology can be included. However, it seems to be clear not only for RSV, that the age of the infected individuals plays a major role in the context of immunopathology. The authors should mention the age dependency and explain how the naïve, mother-derived, young, normal adult and aged immune systems influence the outcome of immunopathology in humans.

The authors are also encouraged to include aspects of the deposition of antigen-antibody immunocomplexes, which are published in several studies and are known for the induction of pathological disorders in the lung of infected individuals.  

In addition, the aspect of sub-optimal antibody binding and low affinity anti-RSV antibodies, which are known to mediate enhanced disease after FI-RSV immunization, should be mentioned.

As a conclusion for the animal model section (before chapter 3.4; line 607), the authors should summarize the most valid animal models for analyzing the immunopathology induced by RSV and also using RSV as a challenge model for vaccine development.

Minor points:

The authors are encouraged to stay at one definition and one abbreviation in the whole manuscript, especially for the CTLs and CD8+ T cells. Please use only one description.

Section 2.11. Please write “CD4+ T cells” (line 387)

Please include a short explanation for the CD4+ T cells and T helper cells, especially before mentioning Th1 and Th2 cells.

Line 79: Please include RSV infection rates during the COVID-19 pandemic. 

Line 551: The authors should mention the most commonly used mouse model based on the BALB/c strain. Different aspects of the RSV immunopathology were simulated and proven by using this mouse strain. Mentioning shortly e.g. the controversial outcomes in the BALB/c mice model would also be beneficial for the presented review.

Line 564: The authors should explain the reasons for this observation of using the F-protein of strain Line 19 to induce higher viral loads in mice, which is important to understand the background results of the cited references.

Line 590: Please correct “For” to “for”.

Line 657: Instead of being unspecific here, please provide the most important marker for the determination of correlates of protection against RSV or RSV-mediated severe disease. In addition, the details on correlates of protection could also be summarized in a short table, which would be beneficial for the fast reading of the manuscript.

In conclusion, the review manuscript presented by Bergeron and Tripp is a well-elaborated overview in the field of RSV-induced immunopathology. Therefore, I recommend the manuscript for publication after revision.
